# Are Efficient-Dose Mixtures a Solution to Reduce Fungicide Load and Delay Evolution of Resistance? An Experimental Evolutionary Approach

**DOI:** 10.3390/microorganisms9112324

**Published:** 2021-11-10

**Authors:** Agathe Ballu, Anne Deredec, Anne-Sophie Walker, Florence Carpentier

**Affiliations:** 1Université Paris-Saclay, INRAE, AgroParisTech, UR BIOGER, 78850 Thiverval-Grignon, France; agathe.ballu@inrae.fr (A.B.); anne.deredec@inrae.fr (A.D.); 2Université Paris-Saclay, INRAE, UR MaIAGE, 78350 Jouy-en-Josas, France; 3AgroParisTech, 75005 Paris, France

**Keywords:** experimental evolution, fungicide resistance, selection drivers, generalism, ecological specialization, environmental variation, selection heterogeneity, mixture, dose variation, *Zymoseptoria tritici*

## Abstract

Pesticide resistance poses a critical threat to agriculture, human health and biodiversity. Mixtures of fungicides are recommended and widely used in resistance management strategies. However, the components of the efficiency of such mixtures remain unclear. We performed an experimental evolutionary study on the fungal pathogen *Z. tritici* to determine how mixtures managed resistance. We compared the effect of the continuous use of single active ingredients to that of mixtures, at the minimal dose providing full control of the disease, which we refer to as the “efficient” dose. We found that the performance of efficient-dose mixtures against an initially susceptible population depended strongly on the components of the mixture. Such mixtures were either as durable as the best mixture component used alone, or worse than all components used alone. Moreover, efficient dose mixture regimes probably select for generalist resistance profiles as a result of the combination of selection pressures exerted by the various components and their lower doses. Our results indicate that mixtures should not be considered a universal strategy. Experimental evaluations of specificities for the pathogens targeted, their interactions with fungicides and the interactions between fungicides are crucial for the design of sustainable resistance management strategies.

## 1. Introduction

The widespread use of pesticides and drugs has led to the rapid evolution of resistance, which reduces or even abolishes their efficacy in some situations [1]. Resistance management is therefore crucial to prevent the overuse of pesticides, which would be deleterious to human health and biodiversity, and to maintain sufficient levels of high-quality agricultural production. It is all the more relevant in a context in which the number of new modes of action (MoA) discovered is dwindling and agricultural practices favour the emergence and spread of resistance [2]. Management strategies aim to slow resistance build-up by maximising the heterogeneity of selection pressure. This may involve dose reduction and/or combinations of different MoAs in space and time [3].

Fungicide mixtures (i.e., the combination of two or more fungicides within the same treatment) are the most widely used, studied and recommended strategy for controlling plant pathogens (FRAC recommendations for fungicide mixtures 2010; REX Consortium 2013). The efficacy of such strategies for delaying the development of resistance and maintaining disease control has been demonstrated in both empirical and modelling studies [4,5,6]. The adoption of this strategy is also driven by practical concerns, as many manufacturers offer ready-to-use commercial mixtures including independent MoAs, although it is often possible to design similar tank mixtures with the same active ingredients (AIs) [7]. Finally, one side benefit of mixtures is that they can be used to control multiple pathogens with a single spray (i.e., they broaden the activity spectrum).

Several non-exclusive processes can account for the efficacy of mixtures. First, mixtures expose pathogens simultaneously to several fungicides (i.e., multiple intragenerational killing (REX Consortium 2013)), and the evolution of specific resistance to each of the mixture components (i.e., multiple resistance) is less likely than the evolution of resistance to a single fungicide [8]. Second, according to the established “governing principles” of resistance management, the growth rate of individuals with single resistances (i.e., resistant to one AI) is decreased by the use of mixtures of fungicides [4,9]. The AIs mixed can control both resistant and susceptible strains, resulting in decreases in the growth rates of both resistant and susceptible strains, and a decrease in the selection coefficient, defined as the difference between these growth rates.

Dose reduction can also be used to control resistance; this strategy acts by reducing the growth rate of resistant individuals [4]. Most of the available empirical and theoretical evidence indicates that high doses increase selection once resistance has emerged, although there are counter-examples that can be explained by the convergence of the dose-response curves of resistant and susceptible strains at high doses [10]. During the emergence phase, the effect of dose is highly specific to the interaction between the fungicide and the pathogen, with high doses having either a beneficial or a deleterious influence on resistance. The use of high doses to keep the pathogen population small limits the mutation load but accelerates the selection of any mutations that do emerge [11]. Theoretical studies have shown that, for an overwhelming majority of realistic parameters of fungicide-pathogen combinations, low-dose strategies better limit the emergence of qualitative resistance [11,12].

The combination of mixtures with dose reduction in “efficient-dose mixtures” (i.e., mixtures of reduced doses of AI but providing a similar level of disease control to that provided by these components used alone at their full authorised rate) may decrease the rate at which resistant individuals are selected, thereby increasing fungicide durability [4]. The socio-environmental benefits of reducing the rates of fungicides in mixtures are obvious, but, in practice, commercial mixtures nevertheless include fungicide components at or close to their full rate for use on their own (e.g., in commercial products used on wheat to control septoria leaf blotch; Appendix A). Efficient-dose mixtures are thus rarely used, possibly due to the difficulties of evaluating their potential advantages. First, such mixtures may not display the beneficial effects of high-dose strategies, long advocated as a means of reducing the occurrence of mutations and, particularly, the selection of partially resistant mutants, putative mutational stepping stones to high-level resistance. Second, the efficacy of efficient-dose mixtures may be equivocal because it may depend on the biology of the pathogen (e.g., its ploidy and mode of reproduction [3,13]), fungicide performance [14], the interaction between mixture components (antagonism or synergism; [13,15,16]) and resistance costs [17]. Third, most studies on mixture durability have focused on the evolution of specific resistance to the fungicide considered most at risk of resistance development, rather than the durability of the mixture itself. Finally, the assessment of mixture strategies usually focuses on their performance during the selection phase rather than the emergence phase of resistance dynamics [3,12].

We performed an experimental evolution study to determine how an efficient-dose mixture could be used to manage resistance, with a view to improving comparisons with strategies based on single AIs. In particular, we analysed how mixture components drove the quantitative and qualitative performance of this strategy. We studied *Zymoseptoria tritici*, an ascomycete responsible for septoria leaf blotch (STB), a major disease of winter wheat [18]. STB accounts for up to 70% of fungicide use in Western Europe [19]. Various degrees of resistance to all authorised single-site inhibitors (i.e., with a single biochemical mode of action)—inhibitors of the polymerization of β-tubulin or benzimidazoles, inhibitors of cytochrome *b* of mitochondrial complex III or QoIs, inhibitors of succinate dehydrogenase (a component of mitochondrial complex II of respiration or SDHIs, and inhibitors of sterol 14α-demethylase or DMIs—have been observed in *Z. triciti* in France [20]. Resistance results from mutations affecting the target sites for these four MoAs. Target overexpression has also been demonstrated for DMIs. Overexpression of the MFS1 transporter causes enhanced efflux [21], a generalist mechanism causing multidrug resistance (MDR) affecting all MoAs but with a limited impact on the susceptibility of isolates.

Using an approach previously developed for the study of resistance selection in alternation strategies [22], we observed the evolution of resistance in a haploid yeast-like easily cultured form of a fully susceptible strain of *Z. tritici*. We first compared the rates of resistance evolution under single or mixed fungicide treatments for three AIs with different modes of action applied in amounts resulting in similar efficacy (i.e., ${\mathrm{EC}}_{90}$). We then determined the cross-resistance profiles of the evolved lines, assessing whether the efficacy of fungicide mixtures was counterbalanced by an increase in the occurrence of generalist resistance profiles. Finally, we investigated how the heterogeneity of selection pressure associated with efficient-dose mixtures determined the cross-resistance profiles in evolved strains, relative to strains exposed to a single fungicide at a similarly effective or lower dose.

## 2. Materials and Methods

### 2.1. General Design

The protocol of the experimental evolution was adapted from that of a previous study [22].

The ancestral *Z. tritici* isolate used was IPO323, which is susceptible to all fungicides. Cultures on YPD plates (20 g L^−1^ dextrose, 20 g L^−1^ peptone, 10 g L^−1^ yeast extract, 20 g L^−1^ agar; USBiological, Salem, MA, USA) incubated at 18 °C in the dark for seven days were used to prepare a founding culture in 25 mL liquid YPD (composition as above, but without agar) in a 50 mL Erlenmeyer flask plugged with cotton wool. This primary culture was incubated in similar conditions for seven days, with shaking at 50 rpm, and was used to establish all the other lines.

The various lines were cultured as described above, in 25 mL liquid YPD medium in 50 mL Erlenmeyer flasks. Each fungicide treatment was repeated on four independent populations (i.e., lines). Each Erlenmeyer flask was inoculated with 10^7^ spores (500 µL of the primary culture). Control lines were not treated with fungicides and contained the same amount of solvent as was introduced for the treated lines. Experimental evolution was allowed to occur over seven-day cycles (i.e., roughly six to seven generations per cycle). This cycle duration made it possible to keep cultures in the exponential growth phase (without reaching stationary phase). At the end of each cycle, 2% of the evolved culture was transferred to a new Erlenmeyer flask containing fresh medium. We ensured that population sizes were equivalent at the start of each cycle by mimicking immigration from external populations through the addition of spores from the untreated line to reach a total of 10^7^ spores for each line. OD_405_ was measured at the end of each cycle and used to calculate population size (see [22] for details). Malthusian growth was calculated for each line as previously described [23]:(1)$$m=\ln(\frac{\mathrm{cell}\;\mathrm{density}\;\mathrm{at}\;\mathrm{the}\;\mathrm{end}\;\mathrm{of}\;\mathrm{the}\;\mathrm{cycle},\;\mathrm{day}\;7}{\mathrm{cell}\;\mathrm{density}\;\mathrm{at}\;\mathrm{the}\;\mathrm{beginning}\;\mathrm{of}\;\mathrm{the}\;\mathrm{cycle},\;\mathrm{day}\;0})$$

Spore concentration and Malthusian growth were normalized against the concentration and Malthusian growth, respectively, of the control line.

### 2.2. Selection Regimes and Selection Doses

We designed selection regimes for studies of the influence of three different factors on resistance evolution. First, selection regimes differed in the number of AIs used (from 1 = direct use to 2–3 = mixtures). Second, the AIs were representative of different MoAs: prothioconazole-desthio (P; a DMI), benzovindiflupyr (B; a SDHI) and carbendazim (C; a benzimidazole). Finally, each AI was applied at several concentrations: an efficient dose (no subscript in line names) and reduced doses (indicated by r1 and r2 in line names). All single fungicides were applied at the full efficient dose and at reduced doses, continuously, over the course of the experiment. All combinations of AIs were applied at the full efficient dose. We observed 16 × 4 = 64 independent lines (Table 1). The experiment was conducted over 10 cycles for all but six of the lines (BP, BCP, Br2, Cr1, Cr2, Pr2) for which it was conducted over only nine cycles.

Efficient doses were chosen so that each treatment, whether a mixture or a fungicide alone, exerted a selection pressure of similar intensity on a naive population. Dose-response curves were established for the three AIs: B, C and P. ${\mathrm{EC}}_{90}$ values were established as the fungicide concentration inhibiting 90% of growth relative to untreated lines after seven days. For each selection regime, we used the ${\mathrm{EC}}_{90}$ as the reference dose because it was not possible to determine the MIC (i.e., the minimal inhibitory concentration) experimentally. Fungicide mixtures were prepared with the same proportion of the ${\mathrm{EC}}_{90}$ for each AI, to ensure a similar contribution of each fungicide to overall efficacy. Dose-response curves were also established for each of the three possible pairs of AIs with a range of proportions of the ${\mathrm{EC}}_{90}$ (i.e., from roughly 0.41 to 0.68 times the ${\mathrm{EC}}_{90}$ of each AI; Table 1). Table 1 details the final doses used in the different selection regimes. We calculated their interaction R, as R = ${\mathrm{EC}}_{90}^{\mathrm{theo}}$/${\mathrm{EC}}_{90}^{\mathrm{obs}}$, with the Wadley formula,
(2)$${\mathrm{EC}}_{90}^{\mathrm{theo}}=\frac{1}{\sum_{i\in M}\;f_{i}\;{\mathrm{EC}}_{90}^{i}}$$
where *M* is the mixture of AIs, $f_{i}$ is the proportion of AI *i* in the mixture (calculated from AI concentrations) and ${\mathrm{EC}}_{90}^{i}$ is the ${\mathrm{EC}}_{90}$ of AI *i*. ${\mathrm{EC}}_{90}^{\mathrm{obs}}$ is the sum of AI concentrations in the mixture [24]. By definition, additive interactions were positive. Synergism was considered to occur if R exceeded 1 and negative interactions were considered to result in antagonism if R was lower than 1.

### 2.3. Establishment of Resistance Phenotype Profiles at the End of the Experiment

At the end of the evolution experiment, we performed droplet tests on each of the lines that had gone through nine cycles (i.e., the last cycle common to all lines) of selection, to characterize their resistance profiles.

For each line, four droplets with spore densities adjusted to 10^7^, 10^6^, 10^5^ and 10^4^ spores mL^−1^ were deposited on solid YPD medium to which a discriminatory dose of fungicide had been added in a square Petri dish. The discriminatory doses were validated in preliminary experiments and were designed to prevent the growth of the susceptible ancestral IPO323 isolate but to allow the growth of reference resistant isolates from our collections. The ancestral isolate IPO323 and a negative control were included in each test. Lines evolved under efficient doses were subjected to eight different conditions: the efficient doses of each of the single AIs, the efficient doses of each of the four AI combinations and tolnaftate at 2 mg L^−1^. We used tolnaftate as a marker of generalist resistance. Lines exposed to reduced doses were subjected to the same set of discriminatory doses and to nine additional discriminatory doses, corresponding to the selection dose of each AI in mixtures (Table 1).

Each test was scored according to the rank of the droplet with the lowest concentration of spores allowing growth (e.g., a score of 2 was attributed if growth was observed for both the first and second dilution, but not for the third or fourth spore dilution).

### 2.4. Statistical Analysis

We compared the mean growth of lines over the course of the experiment by one-way ANOVA with line as a factor. Four ANOVAs were performed, one per mixture. Pairwise comparisons between lines were performed with Tukey post-hoc correction. Resistance dynamics analyses were performed with a non-parametric permutation test (10^4^ permutations) for repeated measures, with spore concentration as the dependent variable, selection regime and cycle as explanatory variables and line as a repeated unit of observation. Multiple pairwise *P* values were obtained after Bonferroni correction. The number of selection regimes against which a line was resistant, and its mean resistance score, were calculated as the number and mean of scores strictly greater than zero in its resistance profile, respectively. Linear models were used for the analysis, with the number of resistances modelled with a quasi-Poisson distribution and the mean resistance score modelled with a logGaussian distribution, with the type of selection regime (a single AI or two-or-three-AI mixture) and the selection regime nested within selection regime type as the explanatory variables.

The structuration of the resistance profiles of lines exposed to single AIs or efficient-dose mixtures was represented by a heatmap of the resistance phenotype profiles detected at the end of the experiment, after nine cycles. The Euclidean pairwise distance was used for the hierarchical clustering of these profiles, with dendrograms for the rows and columns. We also performed a principal component analysis (PCA). The effect of dose is represented by three heatmaps of the resistance phenotype profiles of lines exposed to a single fungicide at efficient or reduced doses.

The effects of AI number, alternation partner (C or P) and their interaction with reduced dose exposure (single fungicide or mixture) on tolnaftate resistance score were investigated with a linear model (quasi-Poisson GLM model determined by stepwise variable selection from a Poisson GLM), with exclusion of the lines in which no resistance emerged (i.e., the control lines and B and BP lines).

All analyses and figures were produced with R 4.0.4 and the packages car, emmeans, factoextra, ez, ggplot2, ggpubr, cowplot, gridExtra, Multcomp and FactoMiner.

## 3. Results

### 3.1. Mixture Durability Strongly Depends on Mixture Components

In this experiment, all selection regimes, whether a mixture or a single AI, were designed to have the same efficacy (90% efficacy) relative to the untreated control. The selection doses were therefore fixed at the ${\mathrm{EC}}_{90}$ (hereafter referred to as the “efficient dose”) after the establishment of dose-response curves for each AI and their four possible mixtures. For the CP mixture, the level of interaction was R = 1.22 with the IPO-323 isolate, which is greater than one and, therefore, suggestive of some synergism. R values were below 1 for the other mixtures applied on the same isolate, suggesting antagonism (BC: 0.74, BP: 0.83 and BCP: 0.79) (Table 1). These interactions (synergy or antagonism) were considered non-significant as R < 1.5 for synergy and R > 0.5 for antagonism, according to the criteria proposed in a previous study [24].

We observed the dynamics of *Z. tritici* after experimental evolution in independent lines subjected to treatment with single fungicides or mixtures of fungicides designed to be 90% effective, for three fungicides with different modes of action: benzovindiflupyr (B), carbendazim (C) and prothioconazole-desthio (P) (Figure 1A). Variability was generally low between the four lines exposed to the same treatment. For lines under continuous exposure to a single AI at its efficient dose, resistance emerged first in lines exposed to C and P: the normalised spore concentration (hereafter referred to simply as the spore concentration) of the C and P lines exceeded 20% (double the initial concentration) after five cycles, and resistance was generalised (spore concentration above 90%) after eight and nine cycles for C and P, respectively. For lines exposed to B, no clear emergence of resistance was observed, with spore concentration remaining below 20% after 10 cycles.

The evolution of lines exposed to efficient-dose mixtures was highly heterogeneous. The BP mixture fully delayed resistance, as no resistance emerged in these lines after 10 cycles, as for the B lines. Dynamics differed highly significantly between BP and P (*P* < 1 × 10^−3^) but dynamics between BP and B were similar (*P* = 0.56). The BC mixture had an intermediate performance, significantly different from those of B and C (*P* < 1× 10^−3^ for both), with resistance emerging after six cycles (i.e., one cycle later than for direct exposure to C but before that for direct exposure to B) and a normalised spore concentration that reached 80% by cycle 10, when resistance was generalised in C lines. The CP mixture was not sustainable, as the emergence and generalisation of resistance at cycles 3 and 5, respectively, occurred more rapidly than in lines exposed to C or P alone (emergence of resistance at cycle 5 and generalisation at cycles 8 and 9, respectively) and resistance dynamics differed significantly from those for P and C alone (*P* < 1 × 10^−3^ for both). The three-way mixture (BCP) yielded intermediate results, with resistance emerging and generalising more slowly than in lines exposed to the least durable mixture, CP (but this difference was not significant, *P* = 0.20) although resistance did emerge eventually, by contrast to the BP mixture (*P* < 1 ×10^−3^).

We compared the global increase in resistance, based on cycle-averaged Malthusian growth rates, which produced a similar ranking of these strategies (Figure 1B). The increase in resistance in BC lines was intermediate, significantly higher than that in B lines but lower than that in C lines (*P* < 0.05). The increase in resistance in CP lines was similar to or significantly greater than that in the corresponding single-fungicide lines. The performance of BP lines was not significantly different from that of B lines, which displayed the highest level of resistance durability. BCP lines were intermediate, with a performance not significantly different from that of the two least durable AI treatments.

CP, the least “durable” mixture, was the only mixture to display any evidence of synergism (non-significant) and was applied with an efficient dose lower than the sum of half the efficient doses of each component.

### 3.2. Efficient-Dose Fungicide Mixtures Select for Generalist and/or Multiple Resistance

We determined the phenotypic resistance profile of each population in droplet tests performed at cycle 9 (Figure 2). As expected, the control lines displayed no resistance to any of the fungicide treatments tested in the droplet test. The lines exposed to single fungicides presented contrasting patterns of resistance. Those exposed to C had a unique, narrow resistance profile characterised by strong resistance to C (mean resistance score of 4, i.e., the maximal score) and moderate resistance to the BCP mixture (mean resistance score of 2). By contrast, lines exposed to P had specific profiles in each of the four repeats, suggesting distinct genotypes, all broader than that for lines exposed to C (on average, P lines were resistant to 3.25 of eight discriminatory doses, whereas C lines were resistant to two) and including various degrees of resistance to P and to CP, but also to tolnaftate (for 3 of 4 lines). Tolnaftate resistance is considered an indicator of multidrug resistance due to enhanced efflux in *Z. tritici* [21,25]. Such patterns are consistent with the evolution of multiple and/or generalist resistance mechanisms. Lines exposed to B, in which no resistance had emerged, displayed no resistance in any of the modalities of the droplet test.

The lines exposed to efficient-dose fungicide mixtures in which resistance had emerged (BC, CP and BCP) had broader resistance profiles than those exposed to a single AI, even P. Indeed, they were, on average, resistant to 2.3 times more testing modalities than those exposed to a single AI (*P* < 1 × 10^−4^), but to a lesser extent, with scores 0.8 times lower for selection regimes against which they were resistant. These lines were resistant to their selection mixture, to various degrees, but also to the other three mixtures and to tolnaftate, especially for BCP lines, which had the highest possible score for resistance to tolnaftate. This, again, suggests that multiple and/or generalist resistance was evolving in these lines. However, these lines were not necessarily resistant to the efficient dose of the components of the selection mixture used alone: BC lines were resistant to C but not B; CP lines were mostly resistant to P but remained susceptible to C; and half the BCP population displayed resistance to B and C whereas the other half presented no resistance to any single AI. The lines exposed to BP, in which no resistance had emerged, also displayed no resistance in the droplet tests.

### 3.3. Reduced Doses of Single AIs Still Select for Resistance

As expected, over the course of the experiment, the control of *Z. tritici* was weaker in the lines exposed to reduced doses than in those exposed to the efficient dose of the same fungicide (Figure 3). In particular, resistance to B emerged in populations subjected to treatment with reduced doses of this fungicide, whereas the emergence of such resistance was prevented by use of the efficient dose. For each AI, mean Malthusian growth was significantly greater in reduced-dose lines than in efficient-dose lines (*P* = 0.04 and *P* = 0.003, for P_r_1 and P_r_2, respectively, versus P, and *P* < 1 ×10^−4^, for all pairwise comparisons between efficient and reduced doses of B and C). Surprisingly, C_r_ lines exposed to reduced doses of C (i.e., 0.4 and 0.45 of the efficient dose in the preliminary data), initially displayed a similar level of control to lines exposed to the full efficient dose (Table 1). Nevertheless, control of the fungus was weaker in these lines, as expected, from the second cycle (Figure 3). The greater continuous increase in spore concentration over time cycles indicates that reduced-dose regimes select for resistance, in addition to providing poorer control over fungal populations. However, it was not possible to test the effect of dose reduction on resistance selection, because lines exposed to full or reduced doses were not subject to the same treatment intensity, making it impossible to dissociate resistance selection from growth control.

### 3.4. Reduced Doses of Fungicides Also Select for Generalist Phenotypes

Heatmaps of the phenotypic resistance profiles confirmed that reduced doses of B, C or P selected for resistance (Figure 4). Lines subjected to selection with reduced doses of B or C and more than half of those exposed to reduced doses of P (five of eight) were resistant to the fungicide used for selection at its efficient dose. The resistance profiles selected at reduced doses were broader than or different from those selected at the efficient dose of the same fungicide. For C, the efficient-dose regime selected a unique resistance profile with high resistance to C and moderate resistance to BCP, whereas the reduced-dose regime selected for generally weaker resistance, but with additional resistance to tolnaftate. For P, the efficient-dose regime selected for resistance to P and CP, and also to tolnaftate, in three of four lines. The reduced-dose P regime selected for BP and BCP resistance (except for one line), but only half the lines were resistant to CP or P and all lines were susceptible to tolnaftate. For fungicide B, the reduced-dose regime mostly selected for resistances to B, BP and BCP that we were unable to compare with efficient-dose regime-induced resistance, because no resistance emerged under efficient-dose treatment.

### 3.5. Resistance Profiles Are Determined by the Balance between Selection Heterogeneity and Reduction of the Dose of Single AIs in Efficient-Dose Mixtures

Resistance spectra differed in terms of the number of fungicides for which resistance was detected and the occurrence of these resistances in the replicates of the different selection regimes (Figure 5). The resistance spectrum of BC lines, including six resistances, corresponded almost exactly to the union of the resistance spectra of B_r_ and C_r_ (with an extra resistance to CP and an absent resistance to B). By contrast, the cumulative resistance spectra of B and C included only two resistances. The CP lines had a similar profile, because the CP resistance spectrum included a common resistance to BC and BP observed only for reduced-dose regimens of C and P but not for efficient-dose regimes. The resistance spectrum of BCP lines was also better explained by the spectra of the reduced-dose B, C and P regimes, which contained more resistances to BC, BP and B than the efficient-dose regime spectra.

In PCA of the resistance profiles established for each line, the first axis corresponded principally to resistance to tolnaftate and BCP, and secondarily to resistance to the two-compound mixtures (Figure 6). This first axis showed that efficient-dose mixtures often selected higher intensity generalist resistance. Indeed, to the left of this axis were lines with narrower resistance spectra (i.e., selected with efficient-dose single-AI regimes). Towards the centre of the PCA were lines with low resistance to tolnaftate and BCP (e.g., C_r_1, C_r_2), and, to the right, were lines with higher rates of resistance to tolnaftate and BCP (all treated with effective-dose mixtures). An analysis of the occurrence of tolnaftate resistance revealed a significant effect of mixture on the selection of resistance to this fungicide, with significantly higher scores for two- and three-way mixtures than for the corresponding AIs used alone (*P* = 0.19 and *P* = 0.002, respectively). This analysis also revealed a positive significant effect on the selection of generalist resistance for lines exposed to reduced doses of C (*P* = 0.0059). No negative or highly positive cross-resistance was observed between the different MoAs (i.e., the correlations between scores for different fungicide testing modalities ranged between 0.14 and 0.66; Appendix A).

The generalist resistance profiles selected in efficient-dose mixtures thus result from both the multiplicity of selection pressures exerted by the mixtures and the reduction of the dose of each of their components.

## 4. Discussion

We investigated the effect of efficient-dose mixtures on the emergence and selection of fungicide resistance, by subjecting multiple lines of a susceptible isolate of *Z. tritici* to fungicides representative of three modes of action, applied either singly at the efficient dose or at a fraction of this dose (${\mathrm{EC}}_{50}$), or as two- or three-component mixtures. Indeed, efficient-dose mixture represents a good opportunity to answer the growing social demand, especially in Europe, to reduce pesticide burden in the environment. Efficient-dose applications of single AIs or mixtures resulted in the same treatment efficacy (${\mathrm{EC}}_{90}$). The effect of efficient-dose mixtures on resistance dynamics differed considerably between mixtures, according to their components: such mixtures were either as durable as the best mixture component used alone, or worse than all AIs used alone. Moreover, efficient-dose mixtures favoured generalist resistance phenotype profiles, with all lines subjected to such regimes displaying resistance to all mixtures, but also to tolnaftate, an indicator of multidrug resistance (MDR), a generalist resistance mechanism already described in field strains of *Z. tritici*. The resistance profiles characterised in lines treated with efficient-dose mixtures resulted from the combined selection pressures exerted by each of the components of the mixture at their reduced doses. Indeed, these profiles were similar to the union of profiles obtained after exposure to reduced-doses of the corresponding single AIs, but with higher scores recorded for modalities associated with generalist resistance (i.e., resistance to tolnaftate and mixtures).

The design of this experiment was similar to that used in a previous study [22] using the same AIs but addressing the issue of the sustainability of alternation strategies. Here, the ranking of times to resistance emergence did not reflect the assumed hierarchy of the intrinsic risks of resistance associated with benzimidazoles (high; C), SDHIs (moderate to high; B) and DMIs (moderate; P) [26]. Indeed, resistance emerged first in C lines and later in P lines, but was never selected in B lines. This discrepancy may reflect differences in temperature and humidity between the two evolution experiments, or most probably differences in treatment efficacy (particularly in the use of ${\mathrm{EC}}_{90}$ rather than ${\mathrm{EC}}_{95}$, leading to a substantial difference in the selection doses for B and C). We therefore considered that the lines in this experiment, which evolved in the same environment, were comparable, but we focused our conclusions on the effects of the C and P AIs and did not interpret our results in terms of intrinsic risks.

### 4.1. Mixtures Were No More Durable Than Single Fungicides Applied at the Efficient Dose

We observed highly contrasting resistance dynamics, despite similar initial disease control, depending on the strategy (single or two- or three-way mixtures) and the components of mixtures. Our findings demonstrate that mixture-based strategies do not systematically provide better resistance control than single-fungicide treatments. This result is contrary to the prevailing view and recommendations concerning mixtures [3,4,27]. Indeed, previous studies have reported an ability of mixture-based strategies to delay the emergence [11] and selection [6] of resistance to a high-resistance risk fungicide, increasing the effective life of this fungicide. However, significant differences between this and previous studies may account for the divergent conclusions.

First, we studied efficient-dose mixtures, as suggested in a previous study [28], based on the argument that mixtures could be used at lower doses, and at the minimal dose still giving effective control in particular, to decrease the selection of resistance. So far, almost all the studies on mixtures have considered full-dose mixtures (but see [14] for an exception). These, like most of those used in the field, are based on the redundant killing principle (using two “poisons”, each at lethal dose, to kill a target—i.e., mixing two distinct fungicides each at the full recommended dose), which relies on selection heterogeneity and dose effect. Indeed, in addition to display concomitantly several MoAs, the global dose is increased in comparison with a solo treatment. On the contrary the “efficient-dose” mixtures considered in this paper rely only on selection heterogeneity. By using the lowest dose of mixture controlling the population, they rather display a “complementary killing” than a “redundant killing”. This could explain the reduced performance of our strategies compared with that claimed for mixture in the literature: once resistance to any of the components of the mixture is present, the control induced by the efficient-dose mixtures will presumably be altered, while the dose of the other(s) component(s) is too low to prevent fungal growth. Besides, by exposing all lines to treatments of similar efficacy, we disentangled the effect of complementary killing from any additive or synergistic effects of combinations of AIs. This would not have been possible using half-dose mixtures, as sometimes suggested in order to keep the overall quantity of fungicide used constant. However, in order to reach a similar treatment efficacy, we modified the fraction of the efficient dose of each component. The CP selection regime included the two fungicides, each at 0.4 times their ${\mathrm{EC}}_{90}$, whereas the doses of the other mixtures included components at more than half the ${\mathrm{EC}}_{90}$ of their component (or one third of the dose for BCP). Considering half-doses might have modified the ranking of mixture strategies. For example, the CP selection regime, which was the least sustainable for the efficient-dose mixture (0.4 × ${\mathrm{EC}}_{90}$-dose mixture) would have included higher doses, possibly resulting in greater durability, whereas the other mixtures would have included lower doses, possibly resulting in lower durability.

Second, we used a naive ancestral population, susceptible to all fungicides, whereas most studies have focused on the selection phase of resistance dynamics, i.e., after resistance to at least one of the components has already emerged.

Third, most studies have focused on the evolution of resistance to only one of the components of the mixture, generally the fungicide considered to be at the highest risk of resistance development. Resistance to the other components of the mixture is often assumed to be insignificant, despite its probable contribution to the gradual growth of the population, and generalist mechanisms are neglected. A previous review [3] identified only four papers considering resistance to both components of two-compound mixtures. Our findings can, thus, be interpreted in terms of the overall durability of the mixture, rather than just the effect of the mixture in delaying a specific resistance phenotype. Finally, we performed an experiment in which it was possible to study resistance dynamics without making a priori assumptions about resistance phenotypes or the mechanisms likely to be selected [29,30,31], whereas previous theoretical studies were limited to the consideration of one or a few resistance phenotypes. Our results support the conclusions of the empirical study by Mavroeidi and Shaw [32] suggesting a strong dependence of the benefit of mixtures on the specific combinations of their components, which required experimental demonstration.

### 4.2. Mixtures Favour Generalist Resistance in a Phytopathogenic Fungus

We found that mixtures favoured the selection of broad resistance phenotype profiles, consistent with multiple resistance and/or generalist mechanisms. Indeed, lines evolved under mixture regimes often displayed broader resistance spectra than those exposed to a single AI, with lower resistance intensity, and growth on tolnaftate. As tolnaftate resistance is considered to be an indicator of MDR [25], we assume that generalist resistance was more likely to occur than multiple specific resistances, although we cannot rule out the possibility of such specialist resistance. Indeed, both types of resistance may coexist within an individual or within a population, as previously described [33] in the “bet-hedging” hypothesis, according to which, in an isogenic population, differently specialized phenotypes with fitnesses varying between conditions, may co-exist in a dynamic equilibrium in a heterogeneous environment. Genetic analysis (e.g., of the promoter of the *mfs1* gene, variants of which are associated with MDR in field isolates of *Z. tritici;* [34]) could be performed to determine the resistance structure of evolved populations, although non-target-site resistance could also be acquired by epigenetic mechanisms [35]. MDR might also be determined in our evolved strains by multiple mechanisms different from those already described for *Z. tritici*, as resistance to B was not observed in all isolates resistant to tolnaftate, as observed in field strains [25].

Our findings, indicating that the use of mixtures favours generalist resistance, are consistent with the findings of at least two other studies, [15,36], for herbicide mixtures and another study, [37], on combinations of antibiotics. MDR is an increasing problem worldwide [38]. Greater attention should, therefore, be paid to this trade-off in the design of resistance management strategies, by including considerations relating to the management of non-target site resistance, for example, as suggested in two previous studies on SDHI fungicides [39,40].

### 4.3. Resistance Profiles Are Shaped by Dose Variation and Should Therefore Be Considered in Management Strategies

In resistance management strategies for fungi, the question of dose rate has generally focused on variation in resistance dynamics: the time to resistance emergence or the selection rate [3,6,11,12]. Our experiment did not resolve this debate, because the growth of susceptible and resistant variants was confounded in observations of fungal growth, and because the reduced doses considered here were too low for any realistic description of resistance management strategies with sufficient disease control. However, it did tackle the question of the dose rate from a new standpoint, by considering the qualitative outcome of selection rather than just the dynamics of resistance.

We observed that strains resistant to the efficient dose of B, C or P could be selected with reduced doses of the same fungicides, even for the lines exposed to B, for which resistance never emerged at full dose. This is consistent with previous observations for antibiotics [31,41,42] and herbicides [43]. Indeed, low-dose treatment leads to the higher frequency selection of resistance mutations with a small effect size, resulting in high-level resistance [43].

The presence of specific resistances in lines treated with reduced-dose regimes suggests that dose mitigation also favours selection for generalist mechanisms. Indeed, resistances to tolnaftate and the BCP mixture were found in lines exposed to reduced doses of B and P, respectively, but not in lines treated with full efficient doses of the same fungicides. These results are consistent with those of many previous studies, in domains other than plant pathology, in which low doses have been shown to select for off-target mutations [44,45,46] and for polygenic resistance mechanisms [44,47] more likely to result in multiple or generalist resistance (see [16] for a review).

The selection exerted by reduced doses of fungicides may also shape the resistance profiles of lines exposed to efficient-dose mixtures, which are more similar to the union of resistance profiles of lines exposed to reduced doses of the components of mixture than to the union of resistance profiles for lines exposed to efficient doses. In particular, resistance to tolnaftate was observed in lines exposed to reduced doses of C (but not in lines exposed to the efficient dose) and in all lines exposed to efficient-dose mixtures including C. As highlighted in a previous study [16] on antibiotics, low doses should be considered with caution in resistance strategy management, as they do not prevent resistance and could lead to the evolution of generalist resistance, even in mixtures.

### 4.4. Experimental Evolution: A Useful Tool for Comparing Strategies

The use of an experimental evolution framework made it possible to subject populations to resistance management strategies with various degrees of selection heterogeneity and to compare the performance of different strategies in standardised conditions. In this controlled environment, it was possible to untangle and assess the performance of several drivers of mixture and dose-reduction strategies, which would have been difficult to achieve in field experiments. The observation of selected resistance profiles was also an advantage over model studies. Despite these multiple advantages, the experiment remained tricky to handle, resulting in the study of only a limited number of strategies. Further studies testing other AIs, different dose ranges for fungicides used alone or in mixtures and double the efficient dose are required to consolidate our conclusions, particularly as concerns the effect of dose in mixtures. In terms of applications, a better understanding of the predictive capacities of such experiments (e.g., by relating growth dynamics and resistance profiles to disease control and in-field resistance frequency) is likely to be the key to designing resistance strategies tailored to the intrinsic properties of pathogens and fungicides. Finally, we tested our strategies on naive populations, susceptible to all fungicides. Applying this approach to populations in which initial resistance is present might make it possible to offer farmers additional advice, as contrasting resistance statuses have been reported in monitoring studies [20].

## 5. Conclusions

Our results demonstrate that the use of mixtures cannot be considered a universal strategy for resistance management. At the minimal dose able to control the disease, the use of a mixture against a naive population may decrease durability and increase generalist resistance relative to single fungicide treatments of similar efficacy. However, efficient-dose mixtures, provided that they have appropriate components, could potentially provide disease and resistance control as effective as that achieved with single-fungicide treatments, at a lower environmental and economic cost. It is therefore essential to take into account the specificities of the targeted pathogens, their interactions with fungicides and the interactions between fungicides, as demonstrated here, together with the frequency and type of resistance already present in the population, in the design of sustainable resistance management strategies including reduction of fungicide rates. Sound resistance management remains a key challenge for the development of a more sustainable agriculture. Experimental evolution is a highly promising tool that can help us to achieve this goal, as a useful complement to theoretical studies and field monitoring.

## Figures and Tables

**Figure 1 microorganisms-09-02324-f001:**
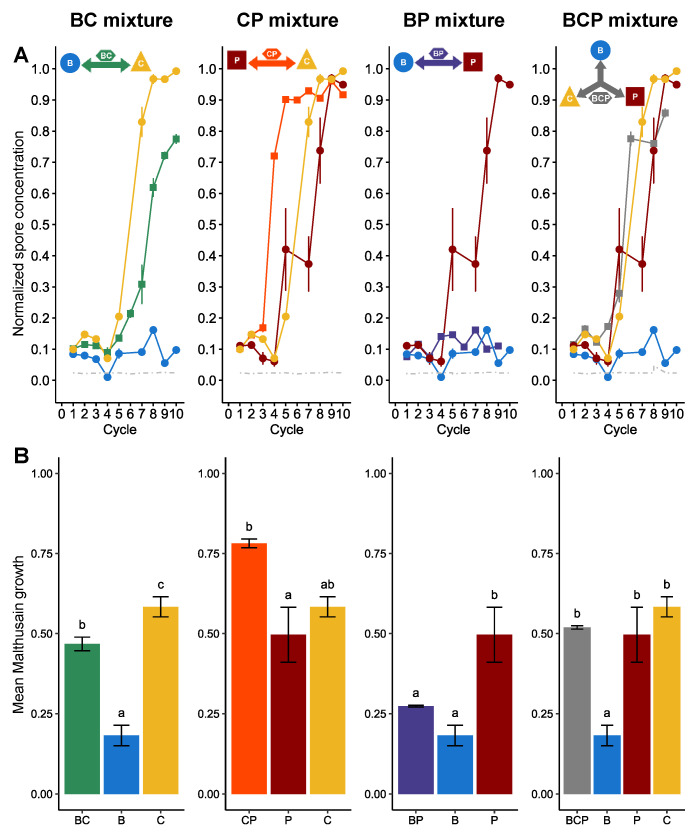
Dynamics of resistance evolution in the lines selected at 90% treatment efficacy. Each column represents the results for a pair of fungicides used alone or as a mixture, at their efficient dose, as explained in the pictograms at the top. B: benzovindiflupyr (SDHI), C: carbendazim (benzimidazole) and P: prothioconazole-desthio (DMI). (**A**) The normalised spore concentration is the spore concentration observed at the end of a cycle relative to that in the control line (i.e., a susceptible population not exposed to fungicides). (**B**) Mean Malthusian growth. Results are normalised against the Malthusian growth of the control (histogram bars) and are presented with their standard deviations (upper and lower lines). Different letters indicate significant differences between groups (*P* < 0.05).

**Figure 2 microorganisms-09-02324-f002:**
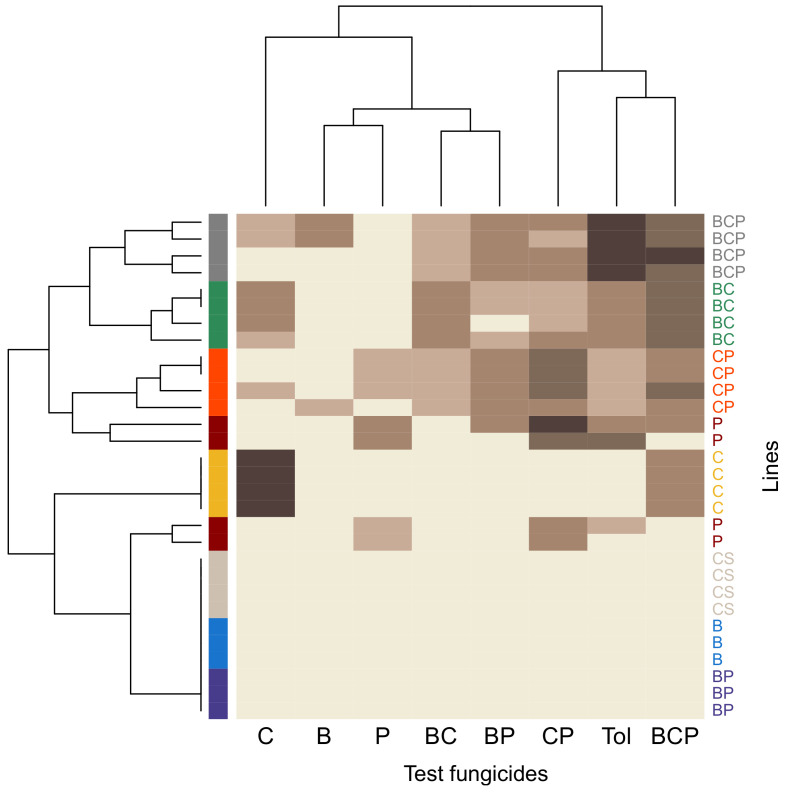
Heatmap of phenotypic resistance profiles at cycle 9. Resistance scores (on the brown scale: from beige = 0 to dark brown = 4) are shown for each of the 12 lines evolved under one of the eight selection regimes (four or three replicate lines per regime; regimes represented by the rainbow scale, as described in Figure 1) and for each fungicide or mixture tested. Names of the lines as in Table 1; CS: control solvent line. Heatmaps were established on the basis of pairwise Euclidean distance.

**Figure 3 microorganisms-09-02324-f003:**
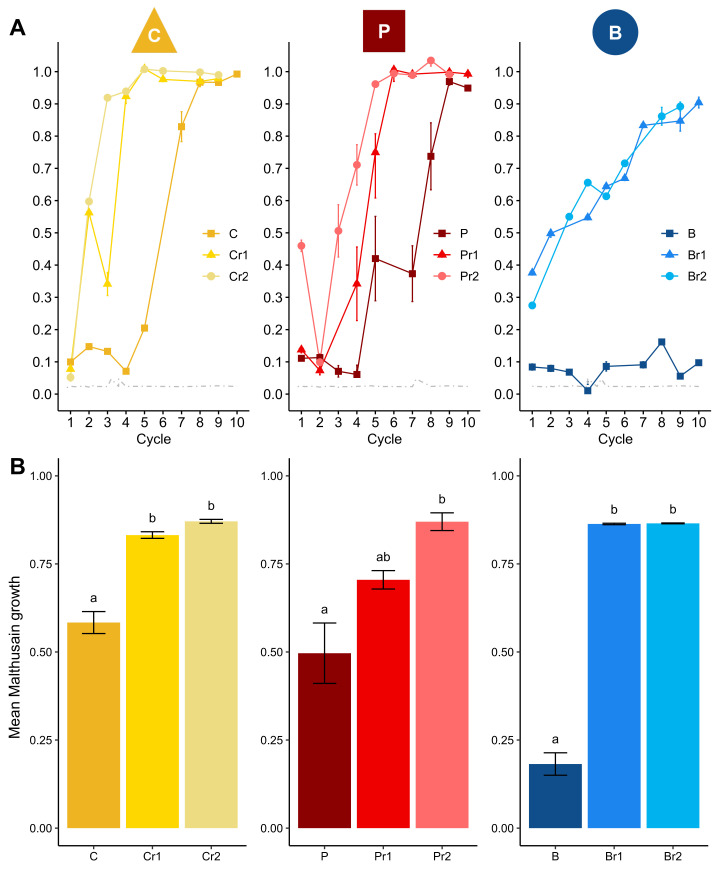
Dynamics of resistance evolution in the lines exposed to a single fungicide at the full efficient dose or a reduced dose. Each column represents results for an AI used at its ${\mathrm{EC}}_{90}$ selection dose or at two reduced doses, corresponding to a fraction of this ${\mathrm{EC}}_{90}$ (Table 1). B: benzovindiflupyr (SDHI), C: carbendazim (benzimidazole) and P: prothioconazole-desthio (DMI). (**A**) The normalised spore concentration is the spore concentration observed at the end of a cycle divided by the spore concentration in the control line (i.e., a susceptible population not exposed to fungicides). (**B**) Mean Malthusian growth. Results are normalised against the Malthusian growth of the control (histogram bars) and are presented with their standard deviations (upper and lower lines). Different letters indicate significant differences between groups (*P* < 0.05).

**Figure 4 microorganisms-09-02324-f004:**
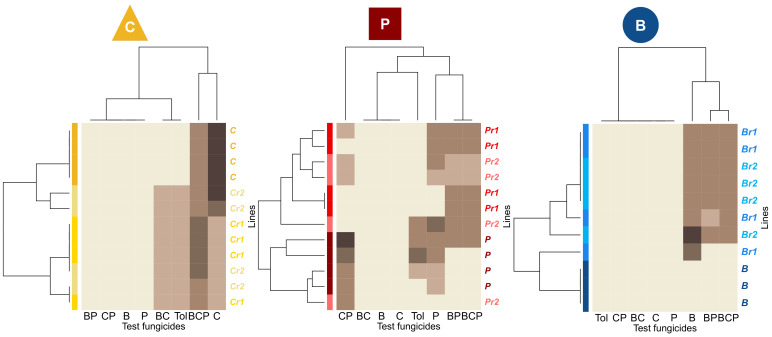
Heatmaps of phenotypic resistance profiles at cycle 9. The resistance rating scores (represented by the brown scale: from beige = 0 to dark brown = 4) are shown for each of the 12 lines evolved under three possible selection doses of single-AI treatments (4 replicate lines per dose) and for each fungicide or mixture tested. From left to right, the single AI used is B (benzovindiflupyr; SDHI), C (carbendazim; benzimidazoles) and P (prothioconazole-desthio; DMI). Heatmaps were established with the pairwise Euclidean distance.

**Figure 5 microorganisms-09-02324-f005:**
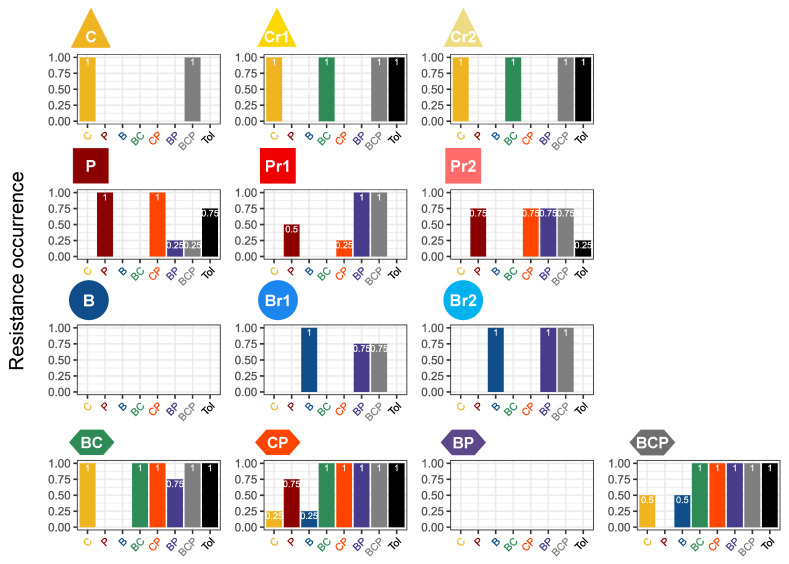
Occurrence of resistance during evolution under each selection regime. The histograms show the occurrence of resistance within a line for each modality in the droplet test. For example, a score of 0.25 means that one of the four replicated lines of this selection regime had a resistance score above zero.

**Figure 6 microorganisms-09-02324-f006:**
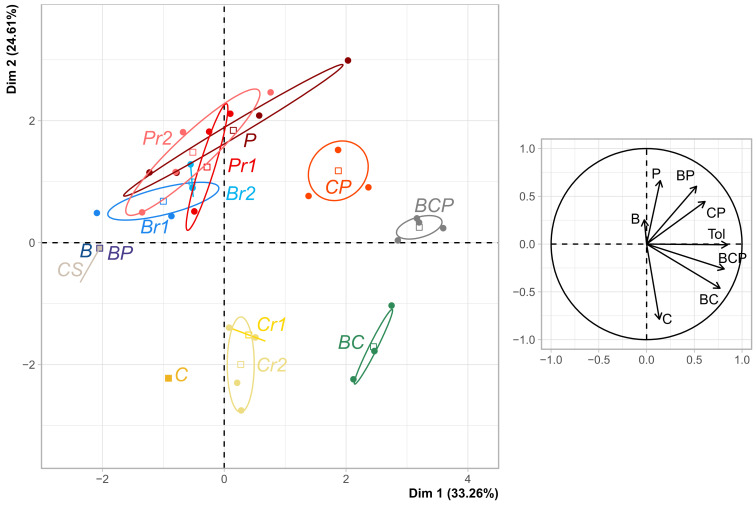
Phenotypic resistance profiles for all lines at the end of the experiment. The PCA was structured by generalist resistance, detected on the basis of resistance to tolnaftate and the BCP mixture.

**Table 1 microorganisms-09-02324-t001:** Doses of fungicides B, C and P and of their mixtures used to select resistance in the various experimental evolution regimes.

Selection Regime	Reference Dose Proportion ^1^	Interaction between Ais ^2^	1st Cycle Efficacy ^3^	B ^4^ (mg L^−1^)	C ^4^ (mg L^−1^)	P ^4^ (mg L^−1^)
**B**	1.00		0.92	0.5		
**Br1**	0.53		0.62	0.263		
**Br2**	0.50		0.73	0.25		
**C**	1.00		0.90		0.2	
**Cr1**	0.45		0.92		0.09	
**Cr2**	0.40		0.95		0.08	
**P**	1.00		0.89			0.005
**Pr1**	0.80		0.86			0.004
**Pr2**	0.60		0.54			0.003
**BC**	0.68	0.74	0.90	0.34	0.136	
**CP**	0.41	1.22	0.90		0.082	0.00205
**BP**	0.60	0.83	0.92	0.3		0.003
**BCP**	0.42	0.79	0.89	0.21	0.084	0.0021

^1^ Proportion of the reference dose applied per cycle. It refers to the efficient dose of the mixture. For example, the selection dose of the CP mixture was ${\mathrm{EC}}_{90}$(CP) = 0.082 mg L^−1^ of C + 0.00205 mg L^−1^ of P, i.e., 0.41 × (${\mathrm{EC}}_{90}$(C) + ${\mathrm{EC}}_{90}$(P)). ^2^ The interaction between AIs was calculated with the Wadley formula [24]. Each selection regime is associated to a specific colour, as used in the results figures, in the first column. ^3^ Efficacy observed at the first cycle. ^4^ B: benzovindiflupyr; C: carbendazim; P: prothioconazole-desthio.

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
