# Peer review of "Are Efficient-Dose Mixtures a Solution to Reduce Fungicide Load and Delay Evolution of Resistance? An Experimental Evolutionary Approach"

_microorganisms, 2021, doi:10.3390/microorganisms9112324_

Round 1

Reviewer 1 Report

This paper provides an interesting take on the debate of the effect of fungicide mixtures and fungicide dose on the selection of resistance and particularly on generalist resistance. The paper reads well and most questions are answered by the time you reach the end of the manuscript. I still have some questions from time to time that I would like the author to answer or comment. I feel that some of those answers might be good to include in the final manuscript but I’m happy with the manuscript has it is and I would like to leave the final decision to the authors. It would have been nice to be able to provide more info on the mechanisms behind the resistance. But this is a great study to show the tremendous impact of MDR and hopefully lead to more knowledge on this obscure and complex resistance pathway. Great work !

Comments:

It is a hard task to relate field dose and laboratory dose (Ec50, MIC,…). The concept of efficient dose might be a good way to bypass this hazardous estimation but how do think it relates to actual field dose ?

L30-35. Consider shortening the sentence

L43-45 I’m not sure that I completely understand this sentence. You imply that farmers use this strategy because commercial fungicide available are formulated with multiple AIs often with belonging to different MoA and It is possible to design mixtures of commercial products that both contain the same actives (thus using the same AI twice in the mixture). If I understood correctly, it seems that there are 2 ideas in the same sentence which makes it hard to read. The first idea is probably more important for your argument.

L 60 indicates “that”

L 128 remove ‘in’

L 149-150 I would remove the “a” in the parenthesis (eg P; DMI)

L 151 maybe replace the content in parenthesis by “ (r1 and r2) “ would be more clear

L 156 I’m not sure I would indicate this justification…

Table 1 Please consider shortening the column name for an easier read. Use footnotes to describe the content of the column rather than doing it in the title.

L 187-190 are discriminatory doses MIC or EC90 ?

Figure 1 Do you have an idea of the R alterations (or combination of alterations) selected in your experiments ? Indeed, selection in a population of a MBC resistance alteration would give it a very high resistance factor to carbendazim. On the other hand selection of a single R alteration in the CYP51 would probably only slightly reduce the efficacy of DMIs. Therefore, being “more resistant than IPO323” to MBC or DMI would not have the same field impact

It seems that mixture “at efficient dose” are weakened by the presence of “high risk” fungicide (here carbendazim) . The mixture without C performs as well as the solo B. Indeed, because of the presence of C in the mixtures, there is less B and/or P (in mg/l). The mixture holds for the first generation but since C is “high risk”, resistance quickly occurs and then there is not enough B and/or P to adequately control the growth. As you mentioned in Table S1, most fungicide are formulated with full rate of each AIs, which is not ideal for the environment but theoretically makes it as if resistance occurs for one component of the mix, there are still enough of the other AIs to control de growth. If we look at the commercial fungicides available on the market, it’s mostly B+P type (SDHI+DMI) formulated with high dose of both AIs. Although modern population of Z.tritici are not naïve, does this mean that this type of mixture would be an efficient resistance management strategy?

Figure 2 Nice Figure. I guess CS is the control lines but I don’t think it is explicitly mentioned in the text or legend. It would be nice to have the legend for each color or at least something like “resistance score (from beige = 0 to dark brown = 4)” in the text legend (also in fig. 4). Would the separation between P line indicates the selection of different alterations ? some lines developed resistance to B even when they were not subjected to B (CP), which indicates a generalist resistance. It is intriguing that in the CP line with B resistance, the Tol resistance was not very high and that in other lines with high Tol resistance, no B resistance was observed… is Tol such a good indicator of MDR/generalism ?

Figure 4 From those result and those of figure 2, it seems like there are different type of “generalist” phenotypes ? Strain subject to C develop a tolerance for Tol and for mixture containing C but never to B or P. The same goes for strain subjected to P. It seems that Tol resistance is not strictly correlated with that but rather that resistance has developed to the subject fungicides and that therefore the mixture containing this fungicide does not sufficiently contain the growth. And with more growth, more development of resistance is possible. On the other hand, one of the Pr2 line is highly resistant to P but fully sensitive to CP and another one is fully sensitive to P, C and Tol and somewhat resistant to CP

Do we have any idea if it is actually related to MFS1 ? Could it be that Tolnaftate resistance, MFS1 promoter inserts and the MDR phenotypes are not strictly related ?

Table S1 good information. I’m not sure if it is because of the review process but the table is split on 3 pages which makes it hard to read

Reviewer 2 Report

This is a well-written manuscript on an important issue in agriculture but examined through an in vitro controlled experiment evolution system. I have only a few comments/suggestions/questions for authors' consideration.

  1. Among the three drugs [benzovindiflupyr (B), carbendazim (C) and prothioconazole-desthio (P)], what are the FICI values in their pairwise combinations (BC, BP, and CP) and all three combination together (BCP) for the original clone? Did the FICI value/direction of interaction change after experimental evolution and selection? Are some combinations synergistic to begin with while others are additive/antagonistic for the initial clone?
  2. How often are the different combinations of fungicides used in agriculture fields? Among isolates from agriculture fields, what are the general patterns of FICI/fungal growth in laboratory media with the different fungicide combinations?
  3. With most drug resistance mechanisms for the three tested fungicides linked to their drug target genes, I believe it's essential that the authors provide DNA sequences of the three target genes for both the original clone and all the evolved clones. Such data will provide additional information and potential explanation on the observed differences among treatments and among repeats within the same treatment. 
  4. In Figure 2, the 4th repeat of CP treatment had elevated growth in the presence of B but not P. Is there a mistake here? If not, what are the potential explanations? Again, sequencing the target gene may offer some clues to this anomaly.

Round 2

Reviewer 2 Report

I think the authors provided satisfactory responses to my comments #1 and #2, their responses to my comments #3 and #4 are unsatisfactory. In the introduction, the authors mentioned that the mechanisms for resistance to the three drugs are well-characterized, primarily at their target genes. Sequencing the three target genes is not an enormous task and would help clarify a number of issues, including comment #4 about potential mislabelling and/or contamination for that outlier observation.

While I agree whole-genome sequencing could provide more information, given the mixed genotypes within each culture, I think it will be more complicated and certainly will take much more time. My suggestion is more limited than that and the targeted sequencing would be more direct and productive to identify the dominant mutations within these cultures. With the relatively few cultures in this study, this is highly doable. If the authors require more time to accomplish this task, the editorial team at Microorganisms should allow them as much time as needed to finish this relatively minor experiment.  
